# On-Chip Antifouling Gel-Integrated Microelectrode Arrays for In Situ High-Resolution Quantification of the Nickel Fraction Available for Bio-Uptake in Natural Waters

**DOI:** 10.3390/molecules28031346

**Published:** 2023-01-31

**Authors:** Sébastien Creffield, Mary-Lou Tercier-Waeber, Tanguy Gressard, Eric Bakker, Nicolas Layglon

**Affiliations:** Department of Inorganic and Analytical Chemistry, University of Geneva, 1211 Geneva 4, Switzerland

**Keywords:** nickel, dynamic Ni-nioxime fraction, integrated antifouling-membrane, adsorptive square wave cathodic stripping voltammetry (Ad-SWCSV), mercury-plated microelectrode arrays, aquatic analysis

## Abstract

We aimed to monitor in situ nickel (Ni(II)) concentrations in aquatic systems in the nanomolar range. To achieve this, we investigated whether an analytical protocol for the direct quantification of cobalt (Co(II)) using adsorptive cathodic sweep voltammetry (Ad-CSV) on antifouling gel-integrated microelectrode arrays (GIME) we recently developed is also suitable for direct Ni(II) quantification. The proposed protocol consists of the reduction of the complex formed between Ni(II) (or Ni(II) and Co(II)) and nioxime adsorbed on the surface of the GIME-sensing element. The GIME enables to (i) avoid fouling, (ii) control the metal complex mass transport and, when interrogated by Ad-CSV, (iii) selectively determine the dynamic (kinetically labile Ni-nioxime) fraction that is potentially bioavailable. The nioxime concentration and pH were optimized. A temperature correction factor was determined. The limit of detection established for 90 s of accumulation time was 0.43 ± 0.06 in freshwater and 0.34 ± 0.02 nM in seawater. The sensor was integrated in a submersible probe in which the nioxime-containing buffer and the sample were mixed automatically. In situ field measurements at high resolution were successfully achieved in Lake Geneva during a diurnal cycle. The determination of the kinetically labile Ni-nioxime fraction allows one to estimate the potential ecotoxicological impact of Ni(II) in Lake Geneva. Additional Ni fractions were measured by ICP-MS and coupled to the in situ Ad-CSV data to determine the temporal Ni(II) speciation.

## 1. Introduction

Trace metals, ubiquitous and persistent compounds, have been exponentially released into the environment since the industrial revolution. Among these, nickel (Ni) is an important bioactive trace metal [1]. In a given range of concentrations, Ni plays a critical role in the assimilation of urea and oxidative defense of phytoplankton [2]. Above or below this optimal concentration range, Ni can cause deleterious effects to phytoplankton [3] and, ultimately, on human health. The nickel concentration in the aquatic environment is regulated by the European Water Directive (WFD-2013/39/EU, 2013; list of priority substances [4]), which specifies that the bioavailable Ni concentration in inland (freshwater) and other (marine and transitional) waters should not surpass the annual averages of 68 and 146 nM, respectively. The Australian and New Zealand governments have established threshold concentrations for hazardous compounds to protect marine species. They reported that available Ni concentrations should not exceed 119 nM to protect 99% of marine species and 136 nM to protect 99% of the freshwater species [5]. The Ni concentration ranges from 2 nM (typical waterborne background) up to 3000 nM in highly impacted aquatic systems [6].

In aquatic environments, Ni exists in different chemical forms or species (speciation). Three defined homologous dissolved fractions can be distinguished by their size: particulate (size ≥ 0.2 µm), colloidal (0.2 µm ≥ size ≤ 1 nm) and the truly dissolved fraction (size < 1nm). The truly dissolved fraction includes the free metal ions and the labile complexes that are potentially bioavailable [7]. The physicochemical conditions, the water composition and various biotic and abiotic processes influence the proportions of these different dissolved forms, which may vary continuously in space and time [8,9,10,11,12]. Therefore, sensing systems allowing for in situ, autonomous measurements of Ni available for bio-uptake at high frequencies are required to improve the Ni ecological risk assessment.

Dissolved trace metals available for bio-uptake in natural waters can be reliably quantified at nM to pM levels by Anodic Stripping Voltammetry (ASV) on on-chip antifouling gel-integrated Ir-based microelectrode arrays (GIME) that are electroplated with appropriate sensing elements [13,14,15,16,17]. Such sensors can be incorporated in submersible sensing probes [11,18] to study in situ or on-board processes that may influence the concentrations of the potentially bioavailable metal species and, therefore, better assess their potential ecotoxicological impact [8,9,10,11,15,16,19,20]. GIME have unique features for the in situ monitoring of complex aquatic ecosystems. Especially, they enable to minimize chemical (fouling) and physical (ill-controlled convection) interferences and to sense homologous dissolved metal species that are available for bio-uptake (for more details, see [21,22]). A wider range of metals can be detected using Adsorptive Cathodic Stripping Voltammetry (Ad-CSV), where the preconcentration of metal complexes is achieved chemically by adding an appropriate ligand to the sample [23,24]. The adaptation of Ad-CSV on GIME, using Hg as the sensing element (Hg-GIME), was recently performed to monitor the dynamic Co-nioxime fraction [25]. Since the nioxime ligand is also selective to Ni(II), we aimed to investigate whether the optimized parameters (nioxime concentration and pH) for the detection of Co(II) are suitable for the detection of Ni(II) and, ultimately, the simultaneous Ni(II) and Co(II) detection in contrasting media compositions (synthetic media, fresh and marine waters). The Hg-GIME was then integrated into a submersible Voltametric In situ Profiling system (VIP [26]), and the developed methodology was successfully applied for the high-resolution in situ quantification of dynamic (potentially bioavailable) Ni-nioxime species in Lake Geneva (Switzerland) during a 24 h measurement cycle. Ancillary measurements of dissolved Ni(II) concentrations in 0.2 µm and 0.02 µm filtered samples were performed by a traditional sampling laboratory ICP-MS analysis. Coupling all these data allowed the determination of the temporal Ni speciation. 

## 2. Results and Discussion

### 2.1. Influence of Nioxime Concentration and pH on the Voltammetric Response

The nioxime or DMG concentration used for Ad-CSV measurements of Ni(II) on bismuth or mercury electroplated macroelectrodes [27,28] is usually equal to or higher than 10^−5^ M, and the pH was fixed to a value higher than 9.2. The study performed for the Ad-SWCSV Co(II) detection on Hg-GIME [25] reflected that the nioxime concentration and pH significant influence, respectively, metal peak currents and peak potentials. Thus, the same protocol was applied here to study the influence of these two parameters on the Ni complexation with nioxime and, therefore, on the Hg-GIME Ad-SWCSV Ni(II) signal and sensitivity. Briefly, the Ni(II) peak current intensities were first recorded in the presence of nioxime concentrations ranging from 10^−8^ up to 5.10^−5^ M in a synthetic media (0.02 M NaNO_3_ and 0.04 M borate buffer containing 12 nM Ni(II) at pH 8.5 to approach the natural pH of fresh and marine waters (see below for more detail) and at pH 9.2. The optimum nioxime concentration was selected and used to study the influence of the pH, ranging typically from 7.46 to 9.25 in both synthetic media (0.02 M NaNO_3_ and 0.04 M borate) and 0.2 µm filtered and UV-digested sea water (UV-SW). 

The Hg-GIME Ad-SWCSV Ni(II) peak current intensities (Δi) measured as a function of the nioxime concentration at pH 8.5 and 9.2 are reported in Figure 1 as a function of the maximum current intensity recorded (Δi_max_). At both pH, the Ni(II) peak current intensities increased with increasing nioxime concentrations up to 10^−6^ M and then remained constant for nioxime concentrations between 10^−6^ and 5.10^−5^ M. A similar trend was observed for Co(II) within the exception of a significant drop in the Co(II) peak current intensities recorded at pH 9.2 for nioxime concentrations between 5.10^−6^ and 1.10^−5^ M. Since a concentration of 10^−5^ M was optimal for both Ni(II) and Co(II) detection at pH 8.5, all further experiments were conducted with this nioxime concentration. For the flow-through cell measurements, a 4.10^−5^ M nioxime solution was stored in a surgical bag, since the nioxime solution was mixed by the VIP multi-channel peristaltic pump with the natural samples at a 1:4 ratio.

Similar to the behavior of Co(II) [25], the peak potential for the reduction of the adsorbed Ni(II) in synthetic media shifted to a more negative potential with increasing the pH (from −952 mV to −1036 mV, Figure 2A). In seawater, the peak potential of Ni(II) was −1072 mV (Figure 2B), independent of the pH. The predicted logarithmic acid dissociation constant (pKa) of nioxime is about 10.70 ± 0.20, while the logarithmic complex formation constant (log*β*_2_) of the 1:2 Ni-nioxime complex is estimated at 17.34 [29]. As explained by Layglon et al. [25] for Co(II), the observed shift of the peak potential in synthetic media suggests an increased complexation of Ni(II) with nioxime with an increasing pH owing to an increased competition for Ni(II) to bind with nioxime relative to hydrogen ions. The absence of a peak potential shift in seawater may suggest a stabilization of the Ni-nioxime complex by chloride, similar to the case of Co(II) but which has not yet been verified.

As observed for Co(II) [25], the peak current intensity increased with the pH up to pH 8.99 and decreased for a 9.25 pH solution (Figure 2A and Figure 3A, left) in synthetic solutions. Visual MINTEQ (ver. 3.1) speciation modeling software predicts that the sum of the inorganic Ni species expected to be dynamic (i.e., free metal ions and nitrate complexes [21]) represents more than 89% of the total Ni(II) for a pH up to 9. It then should decrease owing to the formation of inert hydroxide complexes in a proportion ≥ 20% for pH 9.25 (Figure 3A right). This suggests that the variations of the observed Ni(II) peak current intensities (Figure 2A and Figure 3A left) may be explained in the same way as for Co(II) [25], namely a decrease in hydrogen ion competition up to pH 9 and the formation of Ni-hydroxide complexes above pH 9.2. In 0.2 µm UV-irradiated seawater (UV-SW), the Ni(II) peak current intensity decreased with increasing the pH for the entire range of pH (Figure 2B). As in the case of Co(II) [25], the formation of inert hydroxide complexes alone cannot explain this decrease in signal intensity (Figure 3B right). The precipitation of Ni-carbonate species or adsorption processes on compounds not taken into account in the simulated Ni speciation (e.g., small colloids present in the seawater sample and not influenced by the UV irradiation treatment) may also be responsible for this observation.

These results confirm that, as for the detection of Co(II), a pH of 9.2 usually used in previous reported studies is not suitable. The optimal pH should enable to reach good sensitivity, albeit minimizing the formation of inert Ni-hydroxide complexes. The selected pH should also be as close as possible to the natural pH range, i.e., 7 to 9.5 for freshwaters and 8 to 8.3 for marine waters, to minimize undesired changes in Co and Ni speciation. Taking into account all these criteria, the pH value for measurements in synthetic media and freshwaters was selected at 8.5 and at 8.1 for measurements in seawater, i.e., similar pH values that were selected previously for Co(II) quantification. 

Using these optimal parameters for the nioxime concentration and pH, the simultaneous detection of Co(II) and Ni(II) is now indeed feasible; see Figure 4.

### 2.2. Sensor Sensitivity and Limit of Detection

The sensitivity and detection limits were determined in synthetic and 0.2 µm UV-irradiated fresh and marine samples. The added Ni(II) concentrations ranged from 1 to 18 nM in the synthetic media (Figure 5A) and, respectively, from 6 to 18 nM (Figure 5B) and 3 to 12 nM (Figure 5C) in UV-irradiated fresh and marine samples. Examples of calibration curves obtained under the selected conditions are given in Figure 5. 

Sensitivities and limits of detections (LOD) determined from the linear calibration curves obtained in the various media using a preconcentration time of t = 90 s are summarized in Table 1. A sensitivity of 1.42 ± 0.06 nA nM^−1^ min^−1^ was obtained in the synthetic solution. The sensitivities obtained for the calibrations in UV-irradiated fresh and marine waters were comparable within analytical errors (1.75 ± 0.12 nA nM^−1^ min^−1^ and 1.66 ± 0.15 nA nM^−1^ min^−1^, respectively). The replicates of the calibrations were performed after the renewing of the mercury hemispheres. The relative standard deviations < 10% demonstrate the reliability of the Hg renewal and the repeatability of the measurements. The limits of detection (LOD), calculated as 3× the standard deviation of the intercept divided by the slope of the calibration curve, were below the nM level and comparable within analytical errors for the three media (Table 1). 

Typical Ni(II) concentrations in freshwater are usually lower than 34 nM, while, in seawater, they range from 4 nM in open ocean to tens of nM in coastal areas [1,30]. Therefore, the developed sensor is suitable to determine the Ni(II) concentration available for bio-uptake at a trace level in the aquatic environment and detect anomalies from the anthropogenic release.

### 2.3. Temperature Correction Factor for In Situ Measurements

Since temperature influences the metal diffusion coefficient (D), it influences the metal mass transport of species toward the sensor surface [31]. The temperature of aquatic systems typically ranges from 25 and 4 °C as a function of the depth and seasons. It is required to determine the experimental temperature correction factors to correct the voltametric signals recorded in situ and ultimately insure reliable conversion of these signals into concentrations using calibration slopes determined at room temperature [32]. Previous papers have demonstrated that Equation (1), derived from Arrhenius’ law and the direct proportionality between the peak current intensity and D during the electrochemical preconcentration at microelectrodes (r ≤ 10 µm), may be used for this purpose for cations, neutral and negatively charged oxyanions and negatively charged inorganic metal complexes [13,15,16,17]. Therefore, a linear relationship may also be expected between ln (i) and 1/T (Equation (1)) for the Ni-nioxime complex and the ΔG/R, which is the temperature effect correction factor, experimentally determined from the slope of this linear curve:(1)$$i\propto D\to i=i_{0}\times e^{-\frac{\Delta G}{\mathrm{RT}}}$$
with ∆G the free enthalpy, R the gas constant and T the temperature expressed in Kelvin (K).

Using Equation (2), the experimental temperature effect correction factor can later be applied to correct the influence of temperature on the in situ recorded current:(2)$$i_{T_{\mathrm{room}}}=i_{T_{\mathrm{in}\;\mathrm{situ}}}e^{\frac{\Delta G}{R}\;\left(\frac{1}{T_{\mathrm{room}}}-\frac{1}{T_{\mathrm{in}\;\mathrm{situ}}}\;\right)}$$
where T_in situ_ is the temperature of the sample measured in situ, T_room_ is the temperature used for calibration (20 to 25 °C) and i_Tin situ_ is the current measured in situ with T_in situ_ and i_Troom_ is the corrected current for T_room_. The experiments were performed with a solution containing 20 nM Ni(II) and 10^- 5^ M nioxime and performed in triplicate using linear ramps from 25 °C to 5 °C and 5 °C to 25 °C with steps of 5 °C (Figure 6). The temperature experiments were performed over 10 days without renewal of the Hg, which demonstrates the stability of the mercury hemispheres and reliability of the Hg-GIME.

As expected from the theory, a linear response was observed between ln(i) and 1/T in the three different media of interest (Figure 6). The correction temperature effect factors, ΔG/R expressed in K, for Ni(II) were found to be −3223 ± 229, 3567 ± 143 and 4178 ± 364 in the synthetic media, freshwater and seawater, respectively. The fact that the correction factors in synthetic media and freshwater are comparable indicates that the interactions influencing the measurements are similar in both media. The higher slope in seawater, a more complex medium, suggests that either the same kinetic limiting factor as in the two other media is more affected by the temperature or that another kinetic limiting factor arises from interactions with chloride or other components. The second hypothesis seems more likely. It may be concluded that the peak current behaves as a not entirely reversible system under the fast Ad-SWCSV scan rate conditions used here (0.8 V s^−1^).

### 2.4. Evaluation and Validation of the Analytical Method

To ensure the reliability of the field monitoring data, freshwater and seawater samples were collected, respectively, in La Leyre River and the Arcachon Bay (South-West France), filtered under 0.2 µm and stored in 1 L polyethylene bottle. A first aliquot of the samples was used for Hg-GIME Ad-SWCSV direct dynamic Ni-nioxime quantification (Ni_Dyn_). A second aliquot of the samples was acidified at pH 1. A first subaliquot of the pH 1 sample was analyzed by ICP-MS to quantify the total dissolved Ni concentration (Ni_0.2_), while a second subaliquot was used to assess the Ni_0.2_ measured by Hg-GIME Ad-SWCSV after UV irradiation to decompose the organic matter. After UV irradiation, the pH was adjusted to 8.5 for the freshwater sample and 8.1 for the seawater sample. The Hg-GIME Ad-SWCSV measurements were performed after overnight sample re-equilibration to assess the total dissolved Ni concentrations. The results obtained by ICP-MS and Hg-GIME Ad-SWCSV were then compared.

Ni_0.2_ measured by ICP-MS in both samples were comparable, within analytical errors, to the ones measured by Hg-GIME Ad-SWCSV after UV irradiation (Table 2), which validates our analytical methodology. Ni_Dyn_ determined in the raw samples by Hg-GIME Ad-SWCSV were lower than Ni_0.2_ measured after UV irradiation. The results confirmed that our sensor only detects a small fraction of the total dissolved Ni(II) concentration. They also suggest a low competition of nioxime with nondynamic natural organic and inorganic Ni sorbents. Ni_Dyn_ represented 27% and 13% of Ni_0.2_ measured by ICP-MS (Table 2). 

### 2.5. Field Application in Lake Geneva

During the field campaign, we aimed to quantify in situ both the dynamic (labile and mobile) Ni-nioxime and Co-nioxime fractions (Ni_Dyn_ and Co_Dyn_) defined as the truly dissolved fractions available for bio-uptake. Unfortunately, the Co_Dyn_ concentrations were below the LOD determined for a 90 s preconcentration time, namely lower than 0.29 ± 0.01 nM [25]. The observed temporal concentrations of the three quantified Ni(II) dissolved fractions are shown in Figure 7A. The total dissolved Ni concentrations determined by ICP-MS in the 0.2 and 0.02 µm filtered and acidified samples (Ni_0.2_ and Ni_0.02_) were found to be similar (Figure 7A). Ni_0.2_ and Ni_0.02_ ranged from 6.76 to 8.6 nM and 6.63 to 8.8 nM with mean concentrations of 7.54 ± 0.46 nM and 7.44 ± 0.65 nM, respectively (Figure 7A). The Ni_Dyn_ concentrations were significantly lower during the entire cycle, as could be expected, ranging from 2.59 to 4.33 nM with a mean concentration of 3.33 ± 0.45 nM. These overall data reflect the good status of Lake Leman in terms of the Ni(II) concentrations, i.e., potentially bioavailable Ni(II) concentrations much below the threshold limits (136 nM; ANZEEC, 2000) to protect 99% of the freshwater species.

Hourly variations of Ni_Dyn_ during the entire cycle were up to 67% (Ni_Dyn_ max − Ni_Dyn_ min/Ni_Dyn_ min). This was significantly higher than the hourly variations of Ni_0.2_ and Ni_0.02_, which were up to 27% and 33%, respectively. No correlation was found between the master variables measured (i.e., pH, Eh, O_2_ and Chl*a*). This suggest that the observed highest Ni_Dyn_ temporal variation was probably mainly due to Ni_Dyn_ adsorption/desorption processes on dissolved inorganic and/or organic colloids. This hypothesis is supported by the results of the Ni speciation reported below.

The Hg-GIME Ad-SWCSV and ICP-MS data were coupled to determine the concentrations of three operationally defined specific homologous Ni fractions (Figure 7B), namely: (1) the Ni_Dyn_ fraction potentially bioavailable, obtained by in situ Hg-GIME Ad-SWCSV, (2) the small colloidal Ni species (Ni_SCol_) obtained by subtracting Ni_Dyn_ from Ni_0.02_ and including the metals adsorbed on small inorganic colloids or forming inert complexes with organic ligands and (3) the inorganic and organic coarse colloidal Ni species (Ni_CCol_) obtained by subtracting Ni_0.02_ from Ni_0.2_.

The Ni_CCol_ (Ni associated with coarse colloids) did not represent a large proportion (between 0 and 20%) of the total dissolved Ni, suggesting that coarse colloids did not play a key role in temporal dissolved Ni speciation. The proportion of Ni_SCol_ (Ni associated with small colloids) and Ni_Dyn_ were at the same order of magnitude, ranging from, respectively, 40 to 65% and 34 to 57%. Higher Ni_Dyn_ proportions corresponded usually to the lower Ni_SCol_ ones and vice versa. These opposite trends support the hypothesis of a partitioning of Ni_Dyn_ between the truly dissolved and the small colloidal phases mentioned above. These results reflect the important role that small colloids play on the Ni speciation and especially on the temporal concentration of the dissolved Ni fraction (Ni_Dyn_) available for uptake by microorganisms. The role of small colloids (size < 0.02 µm) on Me available for uptake by microorganisms was already observed and demonstrated for other metals [8,10,11,12,33]. The processes responsible for the partitioning of Me_Dyn_ between the truly dissolved and the small colloidal phases were found to be the photoreduction/oxidation of the small colloids and/or sorption processes issued from change in the pH at the surfaces of microorganisms resulting from photosynthetic–respiration processes [12,33,34]. These processes might also explain the opposite trend observed in this study between Ni_Dyn_ and Ni_SCol_. Of course, more studies are required to verify these hypotheses. 

Overall, these results highlight the requirement of high-resolution spatial and temporal measurements of the potentially bioavailable Ni(II) fraction to better assess its ecotoxicological impact, as it may vary in time independently to the total dissolved fraction.

## 3. Materials and Methods

### 3.1. Chemical and Instrumentation

All solutions used were prepared with ultra-pure deionized water (Milli-Q, 18.2 mΩ·cm, Thermo Fisher scientific Type 1-Ultrapure water system). Boric acid (H_3_BO_3_), nitric acid (HNO_3_) and sodium hydroxide (NaOH) were suprapur grade (Merck, Buchs, Switzerland), while Hg(CH_3_COO)_2_ and KSCN were analytical grade. Standard stock solution of 1 g L^−1^ of Ni(II) and Co(II) (*Trace*CERT^®^, 2% *w*/*w* HNO_3_, Sigma Aldrich, Buchs, Switzerland) and 1,2-Cyclohexanedione dioxime (Nioxime 97%, Alfa Aeser, VWR Nyon, Switzerland), as well as agarose (LGL agarose, Biofinex^®^, Praroman, Switzerland), were sourced from the indicated suppliers. A stock solution of 10 mM nioxime protected from sunlight by aluminum foil was prepared by dissolving nioxime in Milli-Q water and was stored at 4 °C. Standard solutions of Ni were prepared by dilution from the stock solution. 

The influence of temperature was studied using a thermostated glass cell (Metrohm, Herisau, Switzerland) coupled to a Julabo F34 thermostated water bath, with a temperature precision of 1 °C. Sample UV irradiation was performed with a UV digestor (705 UV digester Metrohm).

The total Ni concentrations were measured by ICP-MS (iCAP TQ ICP-MS Thermo^®^, Waltham, MA, USA). Two certified reference materials (CASS-6, nearshore seawater reference material for trace metals, and SLRS-6, river water certified reference material for trace metals; National Research Council Canada) were used as a quality control for the ICP-MS measurements. The recovery was 102 ± 13% and 86 ± 4% for SLRS-6 and CASS-6, respectively.

### 3.2. Design and Preparation of the Hg-GIME

The working electrode used is a gel-integrated Hg-plated Ir-based microelectrode array (Hg-GIME) described previously [13]. Briefly, it consists of an on-chip array of 5×20 Ir microdisks of 5 µm diameter and 150 µm center-to-center spacing surrounded by a 300 µm thick EPON SU8 ring in which a high-purity agarose gel (LGL agarose), acting as an efficient antifouling membrane, is deposited [13]. The array of Ir microdisks acts as a substrate for mercury hemispheres, namely the electrode-sensing element used in this application. The EPON SU8 ring enables a homogeneous, thickness-controlled deposition of the gel and insures the gel mechanical stability [13]. The gel was applied on the surface of the sensor by dipping the chip in a 1.5% LGL agarose solution heated at 80 °C [13,14]. The mercury hemispheres (radius ranging from 5.5 to 6 µm) were electrodeposited throughout the gel by applying a −400 mV deposition potential during 480 s in a 5 mM mercury solution containing 10^−2^ M perchloric acid [13,14]. Thereafter, the Hg was stabilized in 0.1 M NaNO_3_ using Square Wave Anodic Stripping Voltammetry (SWASV) with the following conditions: precleaning potential of −0.1 V for 1 min, preconcentration potential of −1.1 V for 2–5 min and equilibration time of 30 s at −1.1 V. The stripping scan was made from −1.1 V to −0.1 V using a pulse amplitude of 25 mV, a step amplitude of 8 mV and a frequency of 200 Hz. When required, the Hg hemispheres were reoxidized by scanning the potential from −300 to +300 mV in 1 M KSCN for their renewal [13,14]. The gel-integrated Ir-based microdisk arrays were stored in 0.1 M NaNO_3_ between each cycle of measurements. They had a lifetime of typically 1 month [13]. 

### 3.3. Apparatus and Operating Conditions

Electrochemical measurements were performed by using a computer-controlled Voltametric In Situ Profiling (VIP) system ([26], Idronaut S.r.l, Milan; www.idronaut.it). A three-electrode system was used throughout this work. For laboratory measurements, an external homemade plexiglass cell [35] was used with a Metrohm Ag/AgCl/3M KCl/0.1 M NaNO_3_ reference electrode and a Metrohm platinum rod counter electrode. For the final optimization and in situ measurements, an in-house flowthrough plexiglass cell incorporating an Ag/AgCl/ sur-saturated KCl gel mini-reference electrode (Idronaut S.r.l) and a platinum rod mini-counter electrode (Idronaut S.r.l) were used (Figure 8A,B). The VIP encompassed an external double-head peristaltic pump that enables automatic and controlled sample collection, online addition of the nioxime solution (with buffer) contained in a chirurgical bag and their mixing (¼ buffered solution, ¾ natural sample) (Figure 8C). The optimized Ad-SWCSV parameters reported for the detection of the dynamic Co-nioxime fraction [25] were used for the detection of the Ni-nioxime fraction. For completeness, they are given here again in the Appendix A. Briefly, the Ad-SWCSV protocol consisted of the following steps: (1) cleaning of the Hg-GIME at −1200 mV during 30 s, (2) adsorption of the Ni-nioxime complex (or Ni- and Co-nioxime complexes) at −700 mV during 90 s and (3) a stripping scan from −700 mV to −1300 mV to reduce and record the Ad-SWCSV Ni(II) current intensity. Subsequently, a background scan was recorded using the same parameters but with a shortened preconcentration time (10 s). The background scan was then subtracted from the stripping scan to obtain a corrected final analytical scan to improve the signal resolution. The Ad-SWCSV obtained current intensities are therefore referred to as Δi in all the figures. Prior measurements, cleaning of the fluidic system and sample renewal are ensured by renewing 3 times the volume of the fluidic with the natural samples. 

### 3.4. Field Test Area and Sample Collection

Field application and evaluation were performed between 31st August and 1st September 2022 at the LeXPLORE platform at Lake Geneva, Switzerland (Appendix A). Lake Geneva (580 km^2^) is one of the largest freshwater reservoirs (89 km^3^ of water) of Europe currently providing drinking water for approximately 700 000 inhabitants. The LeXPLORE platform is located 570 m from the coast at coordinates 46°30′0.819″ N and 6°39′39.007″ E; see Appendix A. Prior to the submersible probe deployment, the sensor was calibrated on the LeXPLORE platform. The dynamic Ni-nioxime was in situ quantified by Ad-SWSCV every 30 min at a 6 m depth (productive zone). Water samples were collected every hour for complementary measurements of the total dissolved Ni(II) concentrations in 0.2 µm (Ni_0.2_) and 0.02 µm (Ni_0.02_) fractions by ICP-MS. Sampling was performed using an in-house 12V peristaltic pump and acid-precleaned Teflon tubing fixed at the VIP Titanium protective cage. Samples were collected in acid-cleaned 1 L polypropylene (PP) bottles and immediately filtered on-site through 0.2 µm cellulose nitrate filters (Whatman, Basel, Switzerand) and a 0.02 µm alumina-based membrane (Anotop, Whatman, Basel, Switzerland). Filtered samples were collected in 50 mL polyethylene tubes, acidified (0.02% HNO_3_ suprapur, Merck, Buchs, Switzerland) and immediately stored in a cold box. Back to the laboratory, the samples were stored at 4 °C prior to ICP-MS analysis.

To evaluate and validate our Ad-SWCSV procedure, freshwater and seawater samples collected from, respectively, La Leyre River and the Arcachon Bay (South-West France) were filtered under 0.2 µm and stored at pH 1 in a 1L polyethylene bottle. One aliquot of these samples was UV-irradiated at 90 °C during 4 h to decompose the organic matter and assess the total dissolved Ni(II) concentration recorded by Ad-SWCSV after readjustment of the pH at appropriate values (see Section 2.4). A second aliquot of the pH 1 samples was analyzed by ICP-MS. The total dissolved Ni(II) concentrations determined by voltammetry and ICP-MS were then compared.

## 4. Conclusions

This work demonstrated that the optimized parameters for Hg-GIME Ad-SWCSV direct Co(II) detection were suitable for Ni(II), as well as simultaneous Co(II) and Ni(II) detection, knowing the pH value in both freshwater and seawater, as well as the optimal nioxime concentration for the highest sensitivity. Moreover, temperature correction factors were determined for the range from 5 °C to 25 °C for both aforementioned media. The Ni(II) results obtained with the developed methodology and sensor were validated by intercomparison of the total dissolved Ni(II) concentrations obtained by the ICP-MS and Hg-GIME Ad-SWCSV measurements performed in collected fresh and marine samples. These measurements also allowed to evaluate the Hg-GIME labile and mobile Ni-nioxime fractions, defined as the dynamic Ni fraction potentially available for bio-uptake. Finally, after optimization, validation and evaluation, the analytical protocol was successfully applied as part of a field campaign using a VIP submersible probe incorporating an Hg-GIME and an in-house FIA system for the automatic in situ in-line addition of nioxime and a buffer. This work marks the first time that an autonomous electrochemical-sensing probe successfully measured Ni(II) and the hourly temporal variations of its dynamic concentration in situ. The developed system is therefore a promising metal bioavailability-assessment tool that may serve for nickel and cobalt ecological and socioeconomic risk assessments.

## Figures and Tables

**Figure 1 molecules-28-01346-f001:**
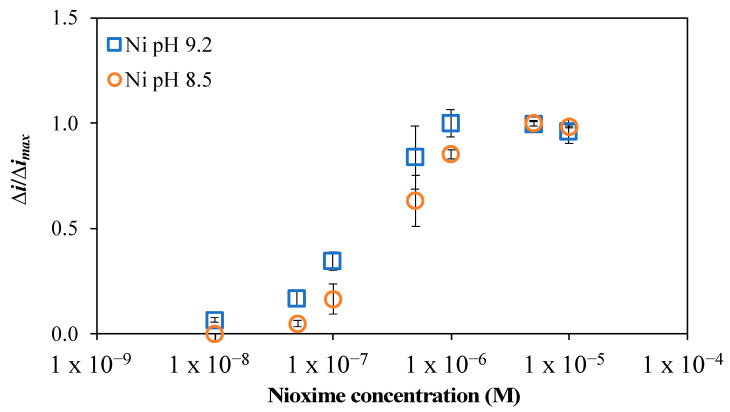
Effect of nioxime concentration on the peak current intensity of 12 nM Ni(II) in a synthetic solution (0.02 M NaNO_3_ and 0.04 M borate) at pH 8.5 (orange open circle) and 9.2 (blue open square).

**Figure 2 molecules-28-01346-f002:**
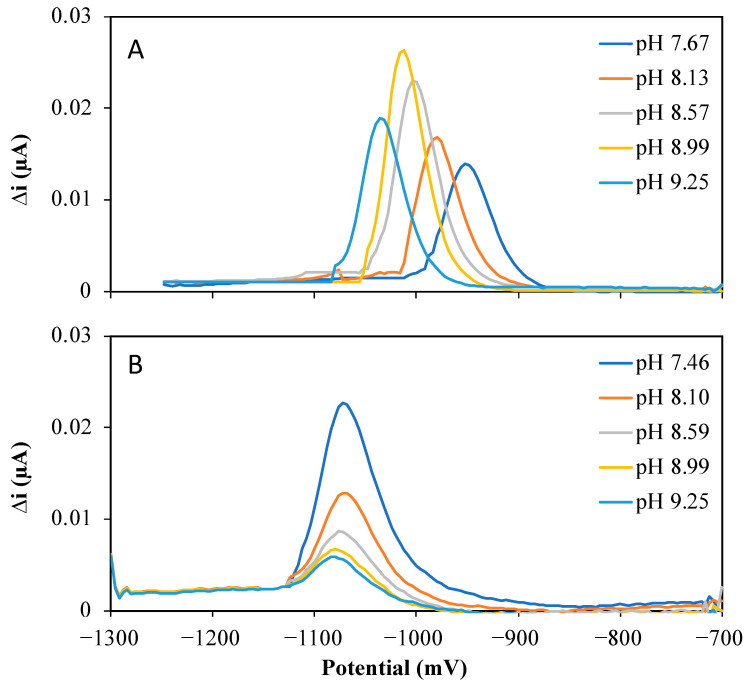
Influence of pH on the Ni(II) peak current intensity and peak potential in both synthetic media (**A**) and UV-SW (**B**). The solutions both contained 10^−5^ M nioxime. The concentration of Ni(II) was 12 nM in the synthetic media and 6 nM in UV-SW.

**Figure 3 molecules-28-01346-f003:**
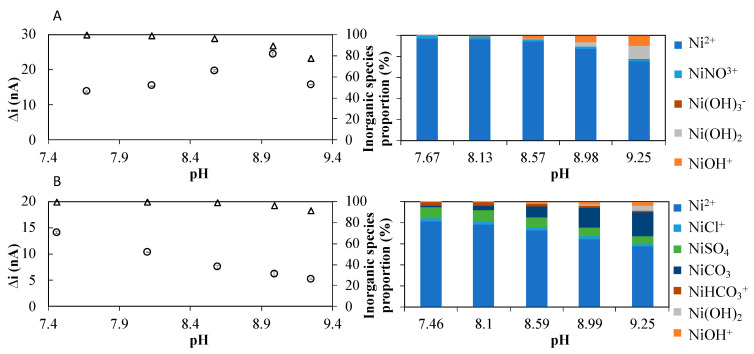
**Left**: Peak current intensity of the Ni recorded by Ad-SWCSV (open circle) as well as the Visual MINTEQ simulated sum of dynamic inorganic Ni species percentage (open triangle) in the range of the studied pH in the synthetic media (**A**) and seawater (**B**). **Right**: Visual MINTEQ predicted inorganic Ni species and their percentage as a function of pH in the synthetic media (**A**) and seawater (**B**).

**Figure 4 molecules-28-01346-f004:**
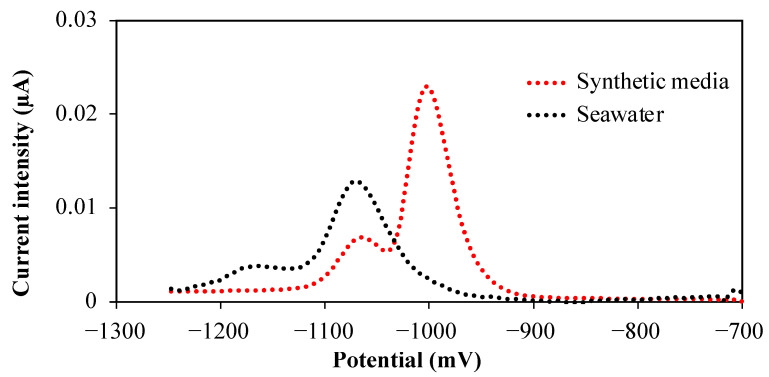
Simultaneous detection of Co(II) and Ni(II) in both synthetic media (red dotted line) and seawater (black dotted line). The added nickel concentrations were 12 nM in the synthetic media and 6 nM in UV-SW, while the Co(II) ones were, respectively, 8 and 2 nM in the synthetic media and in seawater.

**Figure 5 molecules-28-01346-f005:**
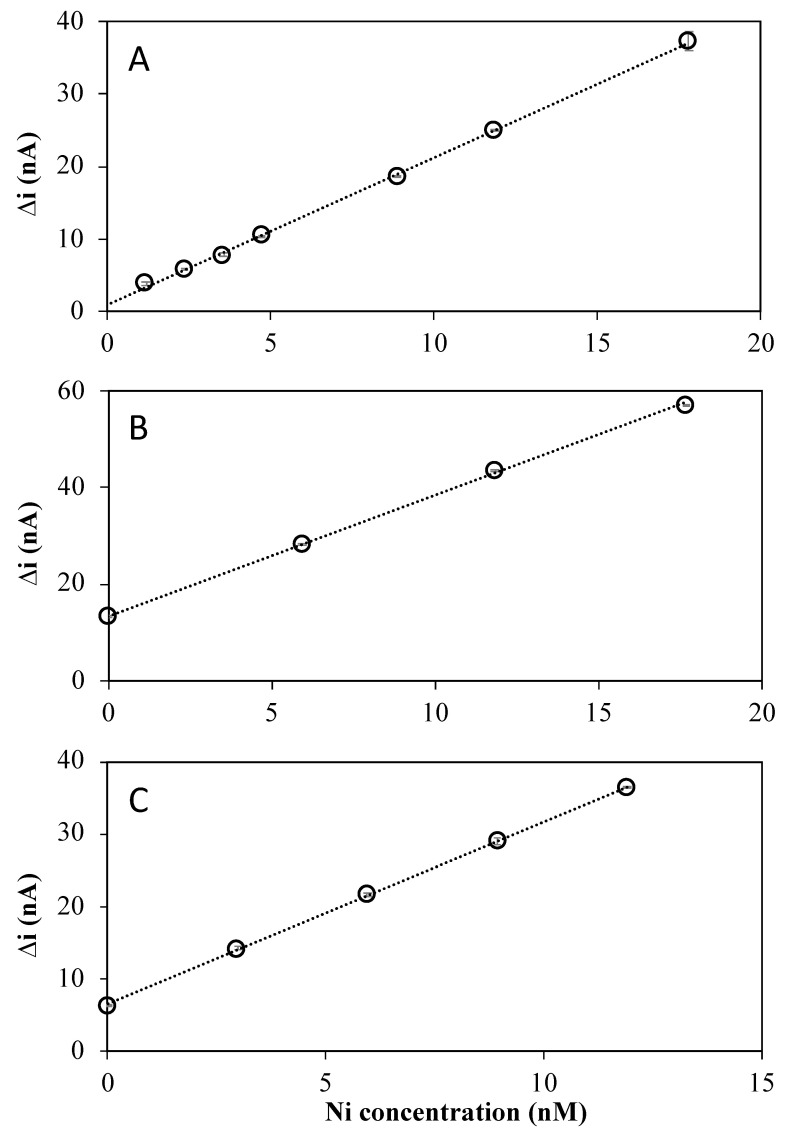
Ni(II) calibration curves in the presence of 10^−5^ M nioxime in (**A**) the synthetic media (0.02 M NaNO_3_ and 0.04 M Borate) at pH 8.5, and UV-irradiated (**B**) freshwater buffered at pH 8.5 (0.04 M Borate) and (**C**) in seawater at pH 8.1 (natural pH).

**Figure 6 molecules-28-01346-f006:**
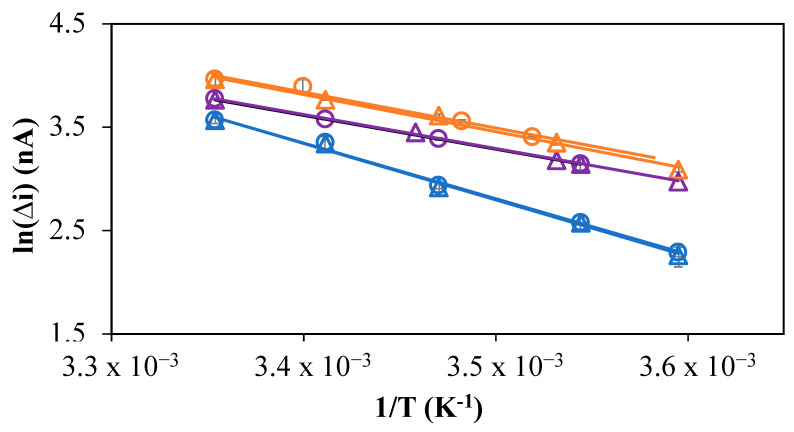
Experimental temperature effect on the Ni(II) peak current intensity in synthetic water (violet), freshwater (orange) and sea water (blue). Each experiment was done from 25 °C down to 5 °C (open circles) and from 5 °C up to 25 °C (open triangles) in triplicate.

**Figure 7 molecules-28-01346-f007:**
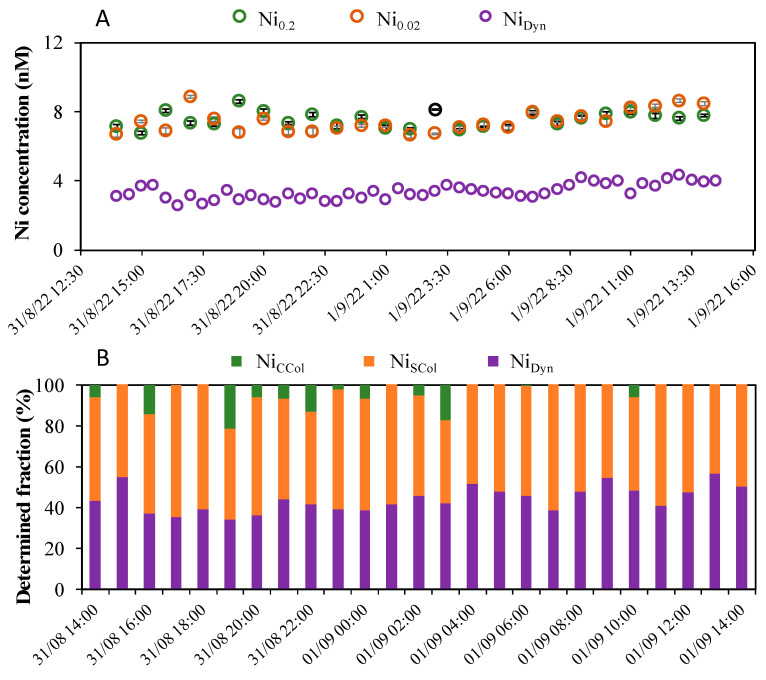
(**A**) Variations of the dissolved Ni(II) concentrations in the filtered 0.2 (green) and 0.02 (orange) µm fractions and the Ni-nioxime concentration (violet), and (**B**) variations of the Ni speciation during the 24h cycle recorded at a 6m depth from LeXPLORE platform in Lake Geneva.

**Figure 8 molecules-28-01346-f008:**
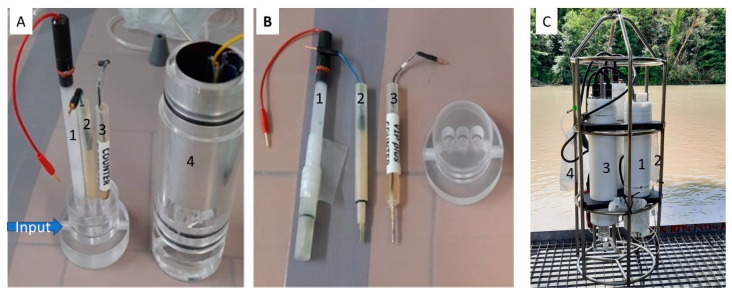
Picture of the (**A**) in-house flowthrough plexiglass cell with the mini-electrodes (1, 2 and 3) and the shielded Plexiglas holder (4) enabling incorporation of the flowthrough cell into the bottom of the VIP electronic housing; (**B**) mini-reference electrode (1), the working microelectrode (2) and the mini-counter electrode (3) and (**C**) the VIP system made of the peristaltic pump (1), the chirurgical bag containing the nioxime and buffer solutions (2); the voltametric probe with the in-house flowthrough plexiglass cell at the bottom (3) and the chirurgical bag to collect the waste (4).

**Table 1 molecules-28-01346-t001:** Obtained sensitivity for Ni(II) in the synthetic media, UV-irradiated freshwater and seawater associated with the calculated limits of detection in nM. Preconcentration time t = 90 s.

			Sensitivity (nA nM^−1^ min^−1^)	LOD 90 s (nM)
Ni(II)	Synthetic media (n = 3)	pH 8.5	1.42 ± 0.06	0.39 ± 0.04
	Freshwater (n = 3)	pH 8.5	1.75 ± 0.12	0.43 ± 0.06
	Seawater (n = 3)	pH8.1	1.66 ± 0.15	0.34 ± 0.02

**Table 2 molecules-28-01346-t002:** Intercomparison of ICP-MS total dissolved Ni(II) and Hg-GIME Ad-SWCSV total dissolved Ni(II) and dynamic Ni-nioxime concentrations.

	ICP-MS Ni_0.2_ (nM)	Hg-GIME Ad-SWCSV		Ni_0.2_ Recovery Ad-SWCSV /ICP-MS (%)	Ni_Dyn_/Ni_0.2_ ICP-MS (%)
		Ni_0.2_ (nM)	Ni_Dyn_ (nM)		
Freshwater	9.22 ± 0.02	10.00 ± 0.75	2.41 ± 0.09	108 ± 8	27
Seawater	7.27 ± 0.25	9.32 ± 0.86	0.94 ± 0.04	128 ± 7	13

## Data Availability

Not applicable.

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
