# Peer review of "On-Chip Antifouling Gel-Integrated Microelectrode Arrays for In Situ High-Resolution Quantification of the Nickel Fraction Available for Bio-Uptake in Natural Waters"

_molecules, 2023, doi:10.3390/molecules28031346_

Round 1
Reviewer 1 Report
The manuscript deals with an analytical protocol that enables direct quantification of Ni(II) and Co(II). The results obtained could be of interest to researchers working in this field.
I suggest to rewrite some parts, especially to rewrite the abstract in order to make clear the purpose of the work and the novelty of the work. I suggest to add some details of the procedure and to improve the comment about the obtaining resulting. You have to better explain the aim of their work in each text to underline the novelty and the performance of the technique.
Please, add the analytical performance of the technique for Ni and Co, for example the recovery on the known concentrations, repeatability....
The quality of the Figures and Tables have to be improved.
I believe the manuscript could be accepted for publication after major revisions.
Abstract
I suggest deleting “Very recently” at the beginning of the abstract
It sounds strange that the title is referred to a method for the determination of nickel and in the first line of the abstract the authors refer to a method for cobalt quantification. Then, in the abstract only details for Ni are reported.
Is “nM-1 min-1” corrected?
Delete “-“ between “90” and “s”
Add a space between “number” and“unit” throughout the text
The authors wrote “Ir based microelectrode arrays”… but some sentences later “Hg-GIME was then integrated…”
2.1 Chemical and instrumentation
No Co-standard solution was cited in the text
The authors modified the electrode with a mercury film, what about its stability?
Please add some information regarding the cleaning step of the system, the duration of the working electrode, in particular considering its applicability for on-site analysis.
Paragraph 2.4.
I suggest to add that “The dynamic Ni-nioxime was in situ quantified BY VOLTAMMETRY” and that an aliquot was stored for the analysis in laboratory by ICP-MS. Is it corrected? Are the results compared?
Is the sentence “The results obtained by voltammetry and ICP-MS were then compared” also referred to these samples? I suppose, yes.
Paragraph 3.1
Please specify “on solid macroelectrodes”, which material? Add references
Please delete “s” from “trends” in “Similar trends was..”
Please specify in the text the which is “Di/Dimax” used in y-axe in Figure 2.
Please specify in the text the which is “Di” used in y-axe in Figures 4 and 6.
Does it make sense to express sensitivity to the minute?
What about the stability of the electrode?
Paragraph 3.4
“The Hg-GIME Ad-SWCSV measurements were performed after overnight sample re-equilibration.” For ICP-MS, were the samples treated with the same procedure?
Avoid the use of “similar” in sentence “The measured concentrations in both the freshwater and seawater samples by ICP-MS were similar”. I suggest to apply a statistical test to demonstrate the agreement between the results obtained by the considered techniques.
Avoid the repetitions throughout the text.
I suggest to better explain the data reported in table 2
Reviewer 2 Report
In this paper, the optimized parameters (especially nioxime concentration and pH) are also suitable for the direct monitoring of Ni(II), and ultimately the simultaneous monitoring of Ni(II) and Co(II), in contrasting media composition (synthetic media, fresh and marine waters) based on antifouling gel-integrated microelectrode arrays were investigated. After reading the manuscript, some issues should be solved.
(1) At part 2.2, the antifouling agarose gel was applied on the surface of the sensor by dipping the chip in a 1.5% LGL agarose solution heated at 80 °C. How to ensure that the LGL does not fall off from the sensor surface? And how to ensure each sensor to be covered with the same thickness of gel layer?
(2) The preconcentration time in this work is 90s. Is 90s the optimum parameter? Why the CoDyn concentrations were below the limit of detection determined for 90-s preconcentration time at the part of Field application in Lake Geneva?
(3) At part 3.5, just some comparison of data cannot prove “These opposite trends support the hypothesis of the partitioning of NiDyn between the truly dissolved and the small colloidal phases” in the manuscript. Please provide more data or references.
(4) As shown in Figure 3A, the peak current of Ni was offset when pH changed. Please make a reasonable explanation for this result.
(5) At part of 3.2, Ni(II) concentrations from 6 to 18 nM and 3 to 12 nM were added to determine the sensitivity and detection limits. But the data in Figure 6 does not match those mentioned in the manuscript. Please check it.
Round 2
Reviewer 1 Report
You have modified the manuscript following my suggestion, improving the quality of data presentation, adding more details, underlying the novelty of the work.
In my opinion, the manuscript can be accepted in the present form
Reviewer 2 Report
I am happy to see that the authors have revised their manuscript carefully according to my suggestion. Now I think it can be published.